



# Multiple limit cycle amplitudes in high fidelity predictions of standstill wind turbine blade vibrations

Christian Grinderslev[1], Niels Nørmark Sørensen[1], Georg Raimund Pirrung[1], and Sergio González Horcas[1]

[1]Department of Wind Energy, Technical University of Denmark, Risø Campus, 4000, Roskilde

**Correspondence:** Christian Grinderslev (cgrinde@dtu.dk)

**Abstract.** In this study, vortex induced vibrations (VIVs) on the IEA10MW blade are investigated using two methodologies in order to assess strengths and weaknesses of the two simulation types. Both fully coupled fluid-structure interaction (FSI) simulations and computational fluid dynamics (CFD) with forced motion of the blade are used and compared. It is found that for the studied cases with high inclination angles, the forced motion simulations succeed in capturing the power injection by the aerodynamics, despite the motion being simplified. From the fully coupled simulations, a dependency of initial conditions of the vibrations was found, showing that cases which are stable if unperturbed, might go into large VIVs if provoked initially by for instance inflow turbulence or turbine operations. Depending on the initial vibration amplitudes, multiple limit cycle levels can be triggered, for the same flow case, due to the non-linearity of the aerodynamics. By fitting a simple damping model for the specific blade and mode shape from FSI simulations, it is also demonstrated that the equilibrium limit cycle amplitudes between power injection and dissipation can be estimated using forced motion simulations, even for the multiple stable vibration cases, with good agreement to fully coupled simulations. Finally, a time series generation from forced motion simulations and the simple damping model is presented, concluding that CFD amplitude sweeps can estimate not only the final limit cycle oscillation amplitude, but also the vibration build-up time series.

**Keywords.** Vortex induced vibrations, fluid-structure interaction, computational fluid dynamics, forced motion simulations, wind turbines, aeroelastic instabilities

**Acronyms**

- BEM - Blade Element Momentum

- BL - Baseline damping

- CFD - Computational Fluid Dynamics

- DTU - Technical University of Denmark

- EUDP - Energy Technology Development and Demonstration Program

- FM - Forced Motion



- FSI - Fluid-Structure Interaction

- IDDES - Improved Delayed Detached Eddy Simulation

- IEA - International Energy Agency

- LCO - Limit Cycle Oscillation

- LES - Large Eddy Simulation

- MBD - Multi-Body Dynamics

- MPI - Message Passing Interface

- NREL - National Renewable Energy Laboratory

- RANS - Reynolds-Averaged Navier-Stokes

- RWT - Reference Wind Turbine

- VIV - Vortex Induced Vibration

- ZD - Zero damping

# 1 Introduction


As wind turbines become larger and more flexible, the risk of serious instabilities between flow and structure increase. An example of these instabilities is the phenomenon known as vortex induced vibrations (VIVs). Here, the shedding of separated flow leads to vibrations of the structure. This does not necessarily lead to any structural issues, however in some cases the so-called lock-in phenomenon can occur. In that situation, the vortex shedding frequency locks in with the structural frequency
in which the blade is vibrating. If this occurs over a sufficient part of the blade span, the corresponding vibrations can become violent and threaten the structural integrity of the blades. The phenomenon of VIVs has been extensively studied for simple geometries such as cylinders, but the number of studies relating to wind turbine blades is quite limited. Main publications of VIVs on wind turbine blades are the ones of Horcas et al. (2019, 2020) and Heinz et al. (2016b), who have studied VIVs for large wind turbine blades (AVATAR10MW (Ceyhan et al., 2016), IEA10MW (Bortolotti et al., 2019) and DTU10MW-RWT
(Bak et al., 2013) respectively). All the aforementioned works were strictly numerical, and relied on fluid-structure interaction (FSI) simulations, involving the communication of a fluid solver and a structural solver during run time, through a loose coupling scheme. These previous studies have identified VIV "risk zones" depending on the angles between flow and blade as well as the velocity magnitude. It is found that the risk of VIVs is very dependent on the so-called pitch $P$ and inclination $I$ angles, as defined in Fig. 1. Assuming that the blade points upwards, the pitch angle $P$ is here defined as the relative horizontal
angle between flow and the root section chord. $P=0°$ when the horizontal wind projection strikes directly the leading edge





of the untwisted blade, $P$=90° when striking the pressure side perpendicularly and by that $P$=180° when striking the trailing edge. $I$ is the relative vertical angle between the inflow wind and the plane intersecting the root section. $I$ is positive when blade points toward the wind with flow from tip to root, and zero when the wind strikes the blade perpendicularly. In particular,

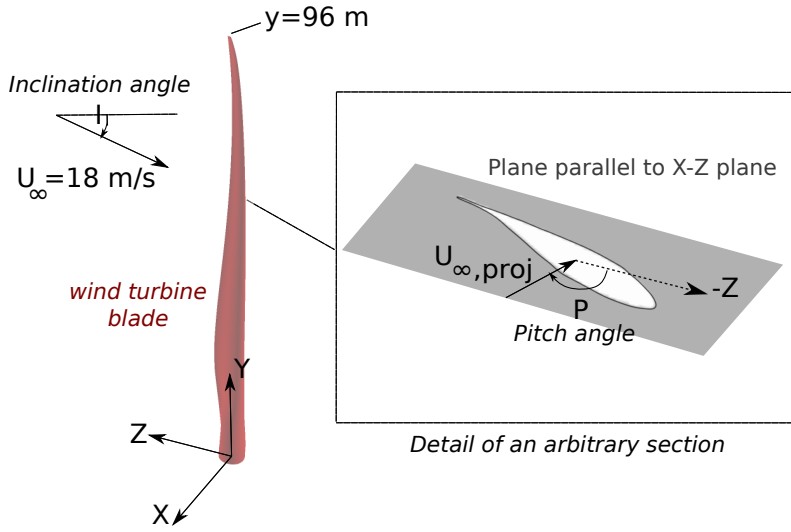

**Figure 1.** Definition of inclination and pitch angles. Reproduced from Horcas et. al (2022) (Horcas et al., 2022).

inclination angles of ≈ 50-60°, and pitch angles of ≈ 80-100° are in the risk zone of high amplitude VIVs for the studied
blade. On top of this, studies have been conducted on the impact of geometry alterations on VIVs such as flap deformations
(Horcas et al., 2019) and alterations of the blade tip (Horcas et al., 2020). These have shown that VIVs are sensitive to the
geometrical alterations, however, these seem to only move the "risk zones" rather than alleviate the phenomenon. In all cases,
the first edgewise vibration mode was found to be excited for these kind of vibrations, at least for the studied conditions. This
mode is likewise the focus on the present study.

The main objective of the present study is the exploration of the forced motion approach as a complementary analysis tool
for the study of wind turbine blade VIVs. The forced motion technique has been extensively applied to academic configurations
such as cylinders (Williamson and Govardhan, 2004). In this approach the surface displacements are imposed, usually through
a simple harmonic function based on the eigenvalues and eigenvectors of the structural problem, rather than being the result
of an aeroelastic interaction computed during run-time (Bertagnolio et al., 2002). For a given inflow, if the imposed deflection
or frequency of motion do not correspond to a physical aeroelastic response, the computed flow and transferred power could
substantially differ from the results of a FSI simulation. For instance, it is well known that the lock-in region becomes narrower
when simulating low amplitudes of motion, both for cylinders (Koopmann, 1967; Blevins, 2001; Placzek et al., 2009) and
airfoils (Meskell and Pellegrino, 2019; Hu et al., 2021). Additionally, the forced motion disregards the structural damping, that
can have an important effect on the VIV phenomenon (Lee and Bernitsas, 2011; Blevins and Coughran, 2009). However, if
assumptions on the loading and on the response can be made, the forced motion strategy is able to estimate the amplitude of



vibrations of a given inflow (Bearman, 2011), which is often the targeted unknown in a design process. This is usually done by performing a series of forced motion simulations of varying amplitude, and subsequently building a reduced order model of the aeroelastic system (Lupi et al., 2021; Zhang et al., 2022). Additionally, the ability to choose the vibration pattern and amplitudes in the forced motion, yields opportunities to investigate the aerodynamics for vibration cases outside those found by

a coupled approach. Another benefit of the forced motion approach is the fact that neither time-marching structural solver nor coupling framework is needed, simplifying the simulations. As the motion of the structure is prescribed, the needed simulation time decreases significantly as no time for the motion build-up is needed.

The current study is part of the PRESTIGE project (EUDP, 2020), looking into stability of wind turbines in various aspects. In this particular work, forced motion and the FSI simulations of the IEA10MW (Bortolotti et al., 2019) blade are performed,

and compared for several conditions. First, simulations are devoted to assess the performance and particularities of both approaches. This initial comparison is followed by two detail studies, aiming at analyzing the impact of (i) the structural damping and (ii) the initial conditions.

## 2 Methodology

### 2.1 Flow solver

To calculate the fluid flow, the computational fluid dynamics (CFD) code EllipSys3D (Michelsen, 1994, 1992; Sørensen, 1995) is used. This code solves the incompressible Navier-Stokes equations through Reynolds Averaged Navier-Stokes (RANS), Large Eddy Simulations (LES) or hybrids of these. In this particular study the IDDES model (Gritskevich et al., 2012) is utilized to effectively resolve the shed vortices, but without having to resolve the viscous sub-layer in LES. The code utilizes structured grids in curvilinear coordinates using the finite volume method. This enables highly scalable computations through

MPI with multi-block decomposition and grid sequencing. Convection is solved through various schemes such as central difference, second order upwind, QUICK or combinations of said schemes. Here, a combination of QUICK and a fourth order central difference scheme is used dependent on the considered region being in a LES or RANS region of the IDDES model (Shur et al., 2008). For the velocity-pressure coupling the SIMPLE algorithm or variations hereof are used along with Rhie-Chow interpolations to avoid odd-even pressure decoupling. Deformations imposed on the CFD surface mesh are propagated

into the volume grid trough a blend factor. This factor is based on the grid line normal distance to the surface and ensures that mesh cells close to the surface are moved as rigid cells, while further from the surface the cells are deformed based on the distance. The solver has been extensively used and validated through multiple studies of wind turbines; see for instance the Mexico project (Bechmann et al., 2011; Sørensen et al., 2016), the Phase VI NREL rotor simulations (Sørensen and Schreck, 2014; Sørensen et al., 2002), and recent comparisons with wind tunnel tests of a curved blade tip (Barlas et al., 2021) and with

the full scale DanAero measurements (Madsen et al., 2018; Grinderslev et al., 2020b; Schepers et al., 2021).




### 2.1.1 Mesh

The structured grid used for the CFD solver is similar to the mesh used in (Horcas et al., 2020), where a sensitivity study of mesh resolution and time step was conducted. It is a body-fitted grid that, for simplicity, only accounts for the geometry of a single blade (and not for the full rotor). The surface mesh consists of 160 cells in spanwise and 256 in chordwise direction (see

Fig. 2). The volume mesh is extruded from the grid surface into a sphere surrounding the blade using the mesh tool HypGrid3D (Sørensen, 1998). The diameter of the sphere corresponds to ≈15 blade lengths. 256 cells are used in the normal direction from the surface with an initial grid size of $1\cdot10^{-6}$ to ensure a $y^+$ of less than 1. The mesh is refined such that cells do not expand rapidly in the region close to the blade where vortices shed from the blade are of interest, and a good cell resolution is needed by the IDDES turbulence model. Further out, the cell size is rapidly expanded toward the boundaries of the sphere. These

boundaries are depicted in Fig. 3, together with a detail of the near-wall volume mesh.

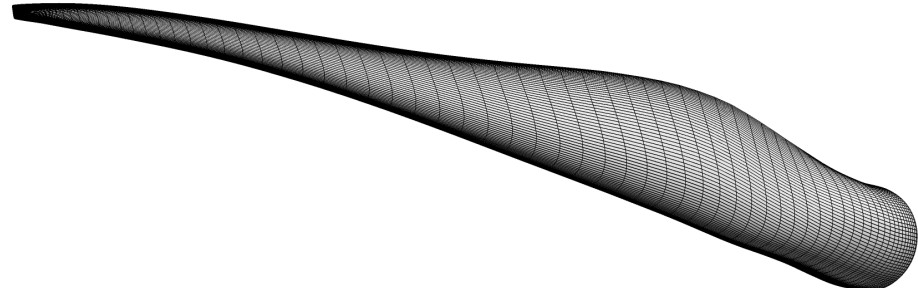

**Figure 2.** Surface grid. Only every second grid line is shown.

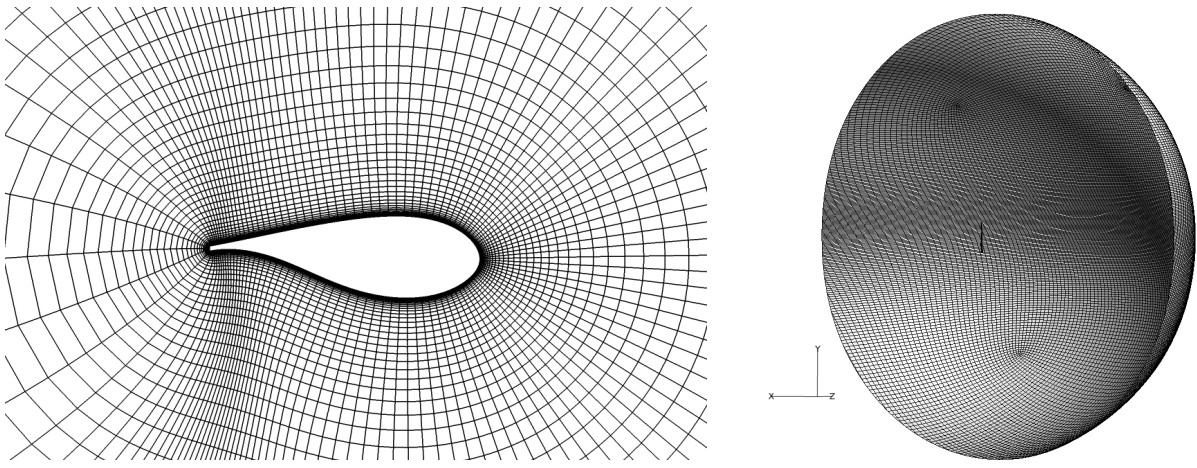

**Figure 3.** Volume grid near surface at 50% blade span and sphere domain (intersected). Only every second grid line is shown.



## 2.2 Coupling framework

To capture the interaction between fluid and structure response, two methodologies are investigated being forced motion (FM), and fully coupled simulations (FSI). These methodologies are also commonly known as 1-way and 2-way coupling, respectively. For both simulation types, it is here for simplicity assumed that the blade is clamped at the root. This means that
singe-blade modes can be captured, but full rotor/turbine modes are omitted.

### 2.2.1 Structural solver

To solve the structural response for 2-way coupling HAWC2 (Larsen and Hansen, 2007; Madsen et al., 2019; Pavese et al., 2015) is used. HAWC2 is a multi-physics code widely used as an aeroelastic-solver using multi-body dynamics (MBD) finite elements for the structure and blade-element momentum (BEM) theory for aerodynamics. Here, only the structural module
is utilized. The blade is represented through a number of finite element Timoshenko beams which are connected through constraint equations to enable considerations of non-linearities. Structural damping is included through Rayleigh damping using the proportionality parameters of the official model of the IEA10MW wind turbine HAWC2 model (Bortolotti et al., 2019). The blade in the present setup is divided into 18 beam elements. Aerodynamic loads are imposed through so-called aero-sections distributed along the blade. In this setup 101 aero-sections, uniformly distributed along the blade, are used.

### 2.2.2 Fully coupled simulations


For the 2-way coupled simulations (FSI), the DTU coupling (Heinz et al., 2016a; García Ramos et al., 2020) is used, which through a Python framework communicates loading and structural response between the flow- and structural solvers. Deformations predicted in HAWC2 are imposed in the EllipSys3D CFD where the corresponding load is found and communicated back to HAWC2 to correct the structural response before moving to the next time step. This is done in a loose coupling, meaning that
only one communication step is done per global time step, while each solver separately uses inner iterations to converge during each time step. Usually, the CFD solver will set the necessary time steps of the simulations, which in the present setup was set to 0.006 seconds per time step, which corresponds to 250 steps per motion cycle for the edgewise mode shape with 0.67Hz frequency. The framework has been used for various studies looking at VIVs (Heinz et al., 2016b; Horcas et al., 2019, 2020) and for wind turbine operation in various flow conditions (Grinderslev et al., 2021a, b), in which the framework is described in
more detail. Additionally, the framework was validated with experimental tests in (Grinderslev et al., 2020a).

### 2.2.3 Forced motion simulations

Forced motion (FM) simulations are conducted directly through the CFD solver, EllipSys3D. This, by imposing the deformation through analytical expressions of the mode shape investigated, being the first edgewise mode. Vibrations are imposed by multiplying the mode shape with a sine function based on the 1st edgewise mode frequency and the amplitude desired at the tip
of the blade. Along with this, a mean deflection of the blade is imposed based on the deflection found from a FSI simulation. The first edgewise modeshape has been fitted through a 7th-order polynomial to the modeshape obtained from the aeroelastic



stability tool HAWCSTAB2 (Hansen, 2004). Both edge- and flapwise components of the first edgewise mode are considered. The torsional component, which amounts to less than half a degree tip torsion for 1 meter edgewise deflection, is omitted. This is possible because the influence of small amounts of torsion on the aerodynamics is negligible at around 90 degrees angle

of attack. The mode shape polynomials along with the corresponding output from HAWCSTAB2 are shown in Fig. 4 and the static mean flap deflection subtracted from FSI simulations and fitted to a 7th order polonomial are shown in Fig. 5. From earlier work (Horcas et al., 2022; Riva et al., 2022), it is known that, for the low wind speeds experienced here, the structural mode shape obtained in HAWCSTAB2 is very similar to the deflection shape seen in FSI.

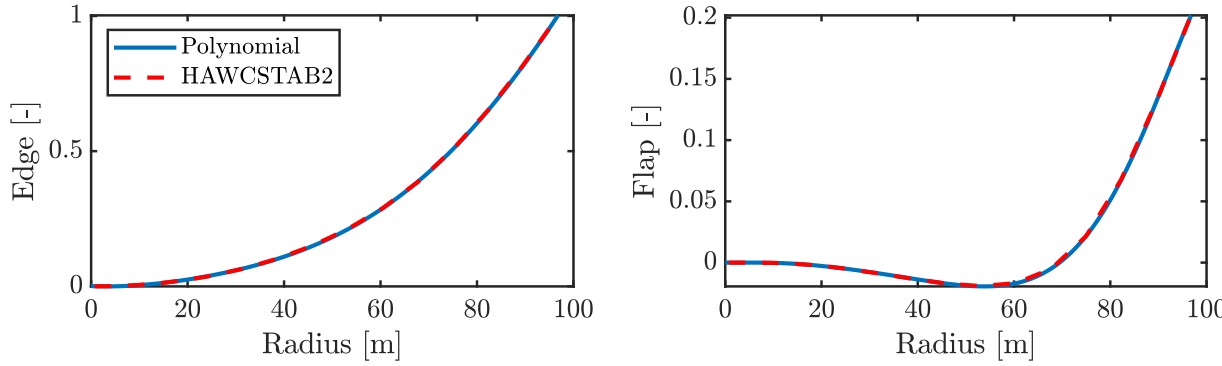

**Figure 4.** 1st edgewise mode shape of blade from HAWCSTAB2 along with polynomial fit used in forced motion simulations.

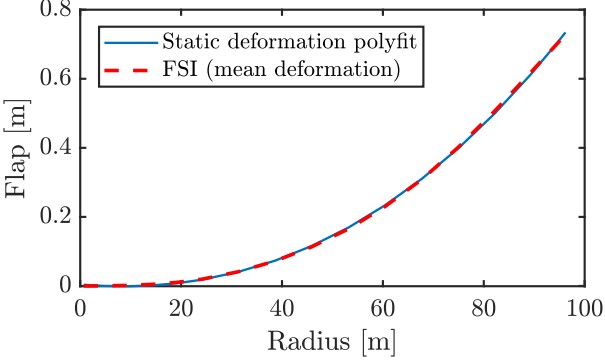

**Figure 5.** Static deflection imposed in forced motion simulations and mean deformation extracted from FSI simulation with flow case *P100-I50*.

## 2.3   Cases studied

As the computations presented here are computationally heavy, three flow scenarios are picked with basis in the study of Horcas et al. (2022). One case which shows high vibrations in the mapping study of Horcas et al. (2022) being the P=100°



I=50° case. Two cases at the vicinity of this case being P=95° and P=90° both with I=50°, which both show no vibration in the aforementioned mapping are also included. These three cases will be identified as *P90-I50*, *P95-I50* and *P100-I50* in the remainder of this paper. All cases are simulated with a uniform inflow of 18m/s, and the angles are obtained by varying the inflow direction. By this, the blade is always pointing upwards with gravity pulling from tip to root. In this study, positive aerodynamic power is defined as injection of power into the structure, meaning that a positive value will increase the vibration amplitudes. The average aerodynamic power $P_{AERO}$ is found by the time averaged cell-wise integration of the power along each of the cells of the blade surface mesh.

$$P_{AERO} = \frac{1}{T \cdot n_{\text{cycles}}} \int_{t=t_0}^{t_0+T \cdot n_{\text{cycles}}} \sum_{i=1}^{N} F_i \dot{u}_i \, dt, \tag{1}$$

with $T$ being the time for one motion period, $i$ is the cell index, $N$ the total number of cells and $F_i$ and $\dot{u}_i$ being the force and velocity at every cell.

## 3   Results

The three aforementioned flow scenarios have been tested for various forced motion amplitudes to investigate the impact on the aerodynamic power injection. In Fig. 6, the resulting aerodynamic power depending on amplitude is depicted for all three cases. As seen, the power in situation *P100-I50*, where vibrations were found using FSI in previous work, follows a quite smooth bell shape with positive power injection from small amplitudes until reaching an amplitude of 1.6m. The blade would reach a limit cycle oscillation at this amplitude if no structural damping is considered. For the two other cases, which were found to be stable in FSI simulations, the initial slope of the power curves is much flatter giving rise to very little power injection in the first half meter amplitude. Therefore, structural damping will give rise to the blade being stable or having only low amplitude vibrations. Going further up in amplitude, the power injection increases rapidly, showing that a range of more than one meter actually has significant positive power injection, which could be higher than what is dissipated by the structural damping. The maximum amplitude with positive power injection is close to the former case around 1.6m.

Case *P100-I50* was further tested in the coupled FSI framework with the baseline structural damping and without any structural damping. This leads to limit cycle oscillations (LCOs) in the edgewise direction, as presented in Fig. 7. As can be seen in the figure, both cases develop LCOs after some time, but stabilizing at different amplitudes, as expected. The undamped case obtains an amplitude of approximately 1.63m, which corresponds well with the intersection with 0W in aerodynamic power injection found for the forced motion simulation presented in Fig. 6. The structurally damped case leads to amplitudes of ≈1.18m amplitude, meaning that the power dissipated by structural damping reduces the vibration amplitude by ≈ 27% for the studied case. If vibration magnitude is the desired outcome of a simulation, the figure above shows a limiting factor for the FSI, as the development time of the LCO is 60 simulated seconds for the baseline case, and even longer for the undamped case.

A clear benefit of the forced motion methodology is that the limit cycle amplitude is set in advance, meaning that no vibration build-up needs to be simulated. By this, the flow shedding just needs to stabilize with the prescribed motion, which is found to



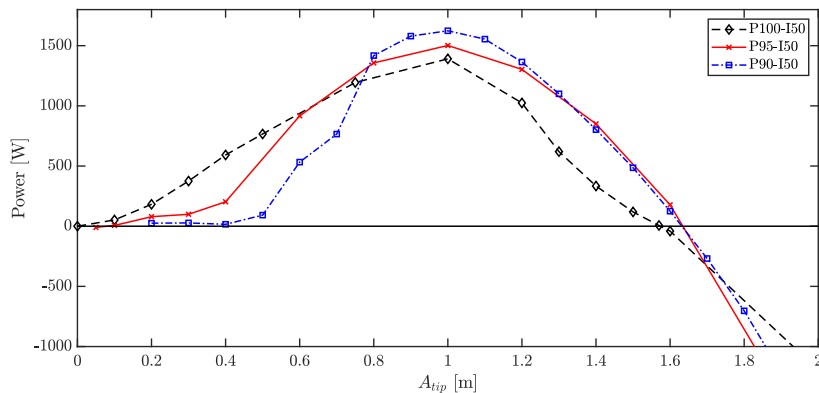

**Figure 6.** Aerodynamic power obtained at various forced amplitudes for flow cases with $50°$ inclination and pitch angles of $90°$, $95°$ and $100°$

.

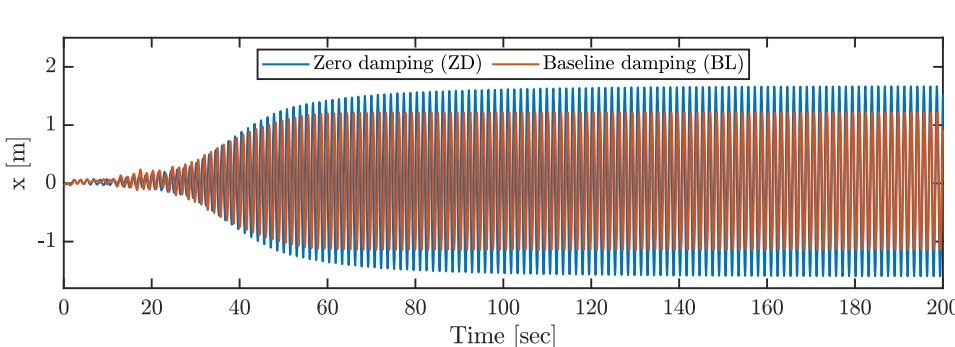

**Figure 7.** Edgewise tip deflection from coupled FSI simulation of flow case *P100-I50* with and without considering structural damping.

only take a few motion periods. This is depicted in Fig. 8, showing the convergence of the total injected aerodynamic power. This benefit, however, diminishes if data for many amplitudes are needed in order to for instance estimate how the vibrations build up.

Taking a close look at the average aerodynamic power injection along the blade span for the coupled and forced motion cases, Fig. 9 shows the results for the flow case *P100-I50*. The spanwise power per cycle $P_{cycle}$ is here found as the injected work averaged over the time span of $n$ cycles as presented in Horcas et al. (2022). The work $w_s$ over $n$ cycles is found as the integration over the time range of the sectional loads $F_s$ and the velocity of the considered section $\dot{u}_s$.

$$P_{cycle} = \frac{w_{s,t_1}}{t_1 - t_0} = \frac{1}{t_1 - t_0} \int\limits_{t=t_0}^{t_1} F_s \dot{u}_s \, dt \tag{2}$$

WIND
ENERGY
SCIENCE
DISCUSSIONS

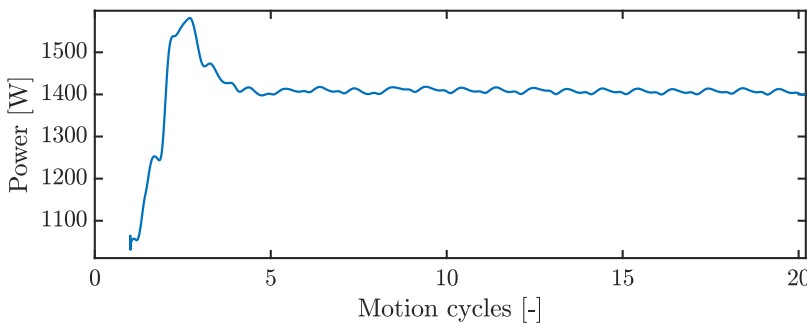

**Figure 8.** Development of aerodynamic power injection (moving average) in FM simulation. Case *P100-I50* with 1.0m amplitude

where $t_0$ is chosen to be after the establishment of the LCO and the integration is performed over $n$ periods of motion:

$$t_1 = t_0 + \frac{n}{f_e} \tag{3}$$

For the FSI simulation results the baseline damping (BL) and a zero structural damping (ZD) case were chosen and compared to the forced motion cases with amplitudes closest to that obtained in FSI, being 1.2m and 1.57m respectively. The general picture shows a very good agreement in both trend and magnitude between the spanwise power injections of the two methods, which indicates that the simplified motion imposed in FM simulations are sufficient. The small differences that occur are likely due to the omission of torsion in FM simulations along with the fact that the amplitudes are not exactly alike between the FSI and FM cases. The main contributor to the total power injection is in the edgewise direction where the motion is largest and the loads interlock with the motion velocity. In the considered cases, especially the range between 40 and 60 meter span of the blade introduces energy to the system, pointing to that range being the critical area where the phase between the flow shedding and blade velocity lock together.

### 3.1 Structural damping fitting

In general, the coupled FSI simulations conducted in this study uses the Rayleigh damping parameters of the official model of the IEA10MW wind turbine HAWC2 model (Bortolotti et al., 2019). However, it is also interesting to asses the effect of other damping coefficients on the VIVs observed and to fit a simple relation between the amplitudes obtained and the power dissipated by damping. A set of simulations of flow case *P100-I50* was conducted through FSI, using various scalings of the structural damping coefficients. A scale factor of $\delta_{\mathrm{DAMP}}$ being [0, 0.5, 1, 2, 5, 10, 15] were used on all Rayleigh damping coefficients for the blades structural model. All cases except for $\delta_{\mathrm{DAMP}}$=15 lead to VIVs, however with varying amplitudes as expected since the equilibrium between aerodynamic power injection and dissipating structural damping will depend on the amplitude. For the high structural damping of 15 times the baseline, the vibrations are removed while small vibrations still occur when having 10 times the baseline damping. Scaling the damping to half or twice the baseline damping yields only marginal effects on the steady state amplitude with an increase (half damp) or decrease (double damp) of $\approx$0.07m (for this reason these are not shown in Fig. 10). This finding is a result of the highly negative gradient of aerodynamic power around the



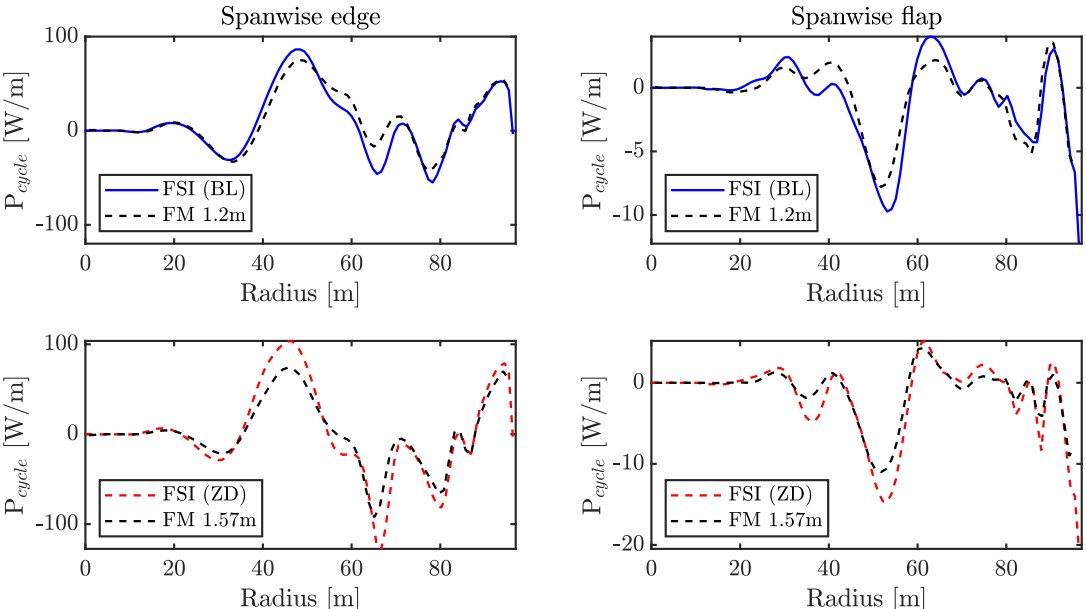

**Figure 9.** Averaged power injected in flapwise and edgewise directions by aerodynamics along span over one motion cycle for FSI and FM simulations. Top row: FSI = baseline damping, FM = 1.2m amplitude. Bottom row: FSI = zero damping, FM = 1.57m amplitude.

seen amplitude, when inspecting Fig. 6. Since the aerodynamic power becomes rapidly higher when lowering the amplitude

only slightly, and vice versa, the doubling of power dissipated structurally will reach an equilibrium with the aerodynamics at only few centimeters less amplitude. The power injected/dissipated is however doubled. Note also, that the development time of the LCOs increase as the damping is increased. This could be beneficial as critical flow conditions need to remain for longer periods in order to develop the VIVs which at the same time become less critical in terms of amplitudes with the increased damping.

A simple model for the power dissipated by structural damping $P_{\mathrm{STRUC}}$ based on tip motion amplitude of the first edgewise mode has been obtained based on the aforementioned FSI simulations with varying structural damping. The power dissipated by the modal structural damping can be computed from the integral of force times velocity, similarly to the power transferred by the aerodynamic forces in Eq. (2). The force due to the modal damping is proportional to the velocity, which means that the dissipated power (force times velocity) is proportional to the velocity squared. The velocity is proportional to the amplitude,

thus the dissipated power due to structural damping is proportional to the amplitude squared. The proportionality constant $F_{\mathrm{STRUC}}$ is here determined from the FSI simulations, where the power dissipated by damping is found using the equilibrium of structurally dissipated power and the aerodynamic power injection when the amplitude of the vibration has converged to a steady state.

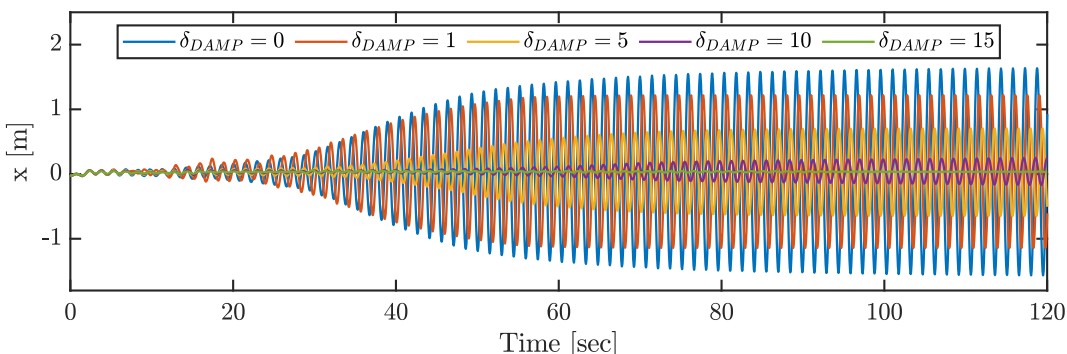

**Figure 10.** Edgewise tip displacements (x) for various scaling of the structural damping Rayleigh parameters.

To test various amplitudes using the coupling, the aforementioned simulations with varying damping coefficients are used. This means that the power dissipated is also scaled accordingly with the same scale $\delta_{\text{DAMP}}$. By this, the fit of a factor $F_{\text{STRUC}}$ can be based on multiple cases. The power dissipated by structural damping $P_{\text{STRUC}}$ can be described as:

$$P_{\text{STRUC}} = F_{\text{STRUC}} \cdot \delta_{\text{DAMP}} \cdot A^2, \tag{4}$$

where $A$ is the tip displacement amplitude in edgewise at steady state vibrations. By the various cases of $\delta_{\text{DAMP}}$ being [0, 0.5, 1, 2, 5, 10, 15], a factor of $F_{\text{STRUC}} = 540 \text{W/m}^2$ is found suitable for all cases as depicted in Fig. 11.

This simple model, only estimates the structural damping dissipation when vibrating in the first edgewise mode as this is what it is fitted for here. Another mode would yield another value of $F_{\text{DAMP}}$ as the motion differs. This model can be used to asses the dissipated power from structural damping for various amplitudes, making it possible to estimate an equilibrium amplitude by use of forced motion simulations with prescribed amplitudes.

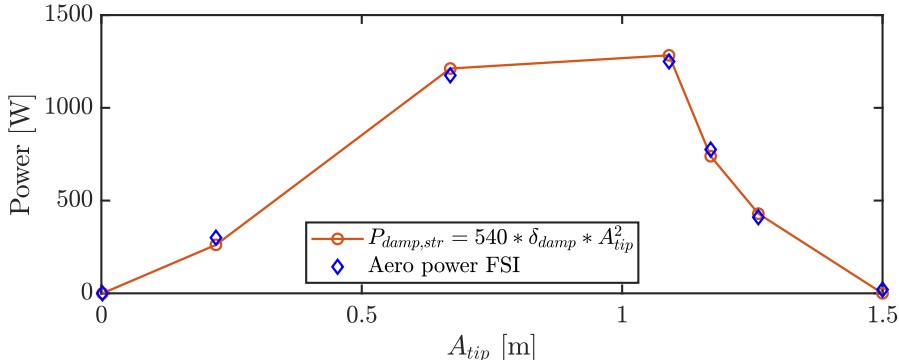

**Figure 11.** Simple damping model of dissipated power compared with aerodynamic power obtained through FSI simulations



### 3.1.1 Amplitude estimations from forced motion simulations

Having obtained a simple relationship between dissipated power and amplitude from the structural damping, the FM simulations can be used to not only assess the amplitude where the aerodynamics stop injecting power, but also assess the limit cycle amplitude when including the structural damping. This amplitude can be found at the intersection between the curve of the structural damping and the aerodynamic power. An example is given in Fig. 12 for flow case *P100-I50* with the FM aero power curve along with the curve of the damping model along with the result of the FSI simulation. As seen, the aerodynamic power

and motion amplitude match well between the coupled FSI simulation and the cross section between the forced motion curve and the damping model. For FSI, the equilibrium state is found at $\approx$1.18m amplitude with an aerodynamic power of $\approx$775W. The equilibrium for the forced motion case and the damping model is found at $\approx$1.24m with a corresponding aerodynamic power of $\approx$880W. This difference in aerodynamic power injection might seem high, but as indicated before, this intersection happens at a very steep range of the power injection curve, which is also seen by the only 6cm difference between the FSI

resulting amplitude and the FM estimated amplitude.

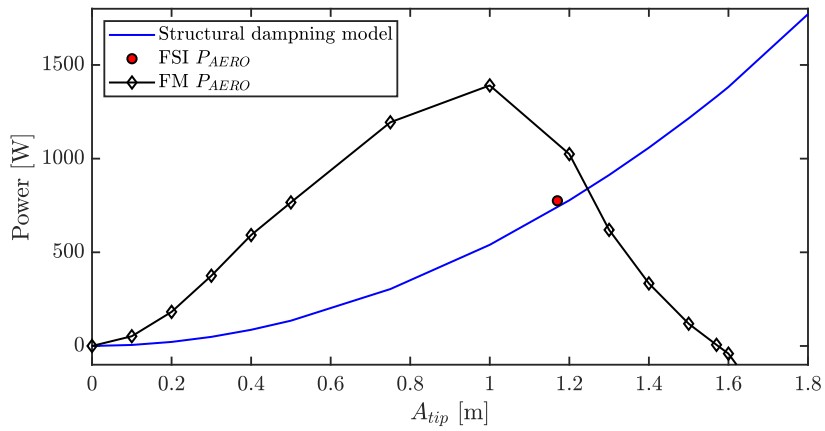

**Figure 12.** Forced motion results for various amplitudes along with FSI result and damping model for baseline structural damping. Case *P100-I50*.

### 3.2 Multiple-amplitude cases / Initial condition dependencies

Originally in the work of Horcas et al. (2022), cases *P90-I50* and *P95-I50* were found to not cause VIVs, as found in the neighbouring case *P100-I50*. However, from the FM amplitude sweeps depicted in Fig. 6, it is seen that a region of positive power injection occurs, when forced amplitudes are higher than $\approx$0.1m-0.4m depending on flow case. This result is also found

through FSI simulations, if initial motions are triggered through an external forcing/motion. Examples of this could be buffeting motion from turbulent gusts or motion from inertial forces when moving the blade through a yaw, pitch or azimuthal change. In this work, various ways of triggering the VIVs branches have been tested, these being turbulent buffeting inflow, a wind





speed ramp and finally externally prescribed synthetic initial forcing of the mode through the FSI coupling. All these methods
are found to successfully trigger the VIVs as the amplitude of vibration merely needs to reach a certain value before the power
injection takes care of building up the VIVs. The latter synthetic trigger method allows most control, and is therefore chosen
in this study. By initially forcing the blade harmonically in the 1st edgewise frequency until an amplitude point is reached, or
merely pulling the blade in the edgewise direction by a static loading and releasing it, triggers VIVs as depicted in Fig. 13.
The figure shows the low equilibrium (LE) LCO being triggered by a harmonic forcing in the first 50 seconds, and the high
equilibrium (HE) being triggered by pull-release. After the period of synthetic forcing, the coupling continues as normal with
the CFD forcing and the structural response calculations until a steady state response is found. The forced triggering of VIVs
also allows a speed-up of 2-way coupled simulations, if the final LCO amplitude is the desired result, since the CFD does not
need to be active in the initial forcing.

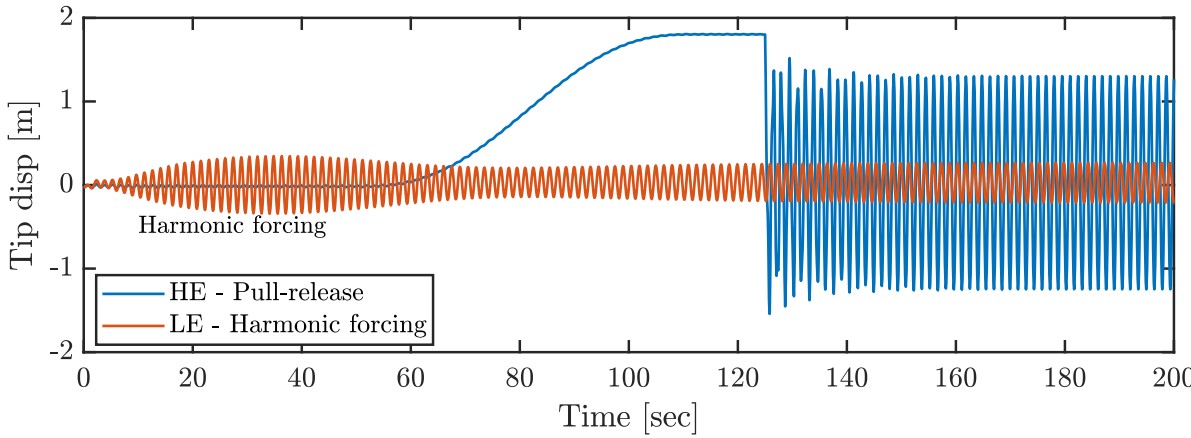

**Figure 13.** High (HE) and low (LE) equilibrium LCOs triggered by synthetic initial conditions for flow case *P90-I50*. HE is triggered by a
pull and release, while LE is triggered by harmonic forcing of the blade for the first 50 seconds.

Considering again the power curves of the three studied flow cases but this time subtracting the power dissipated by the
structural damping, a stability figure can be obtained as presented in Fig. 14. Here, the effective power injection is found as
the difference between aerodynamic power injection and the structural damping dissipation (Eq. (4)). As can be seen, the FM
simulations and the FSI simulations do not agree entirely at low amplitudes, as the FM estimates slightly positive effective
power injection for case *P95-I50* in the whole range of amplitudes from ≈0.1m up to ≈1.4m. In FSI, this case finds three
equilibrium LCOs being ≈0.1m (as found in Horcas et al. (2022)), 0.33m and 1.4m. By use of synthetic initial forcing in the
FSI simulation with varying intensity three resulting limit cycle oscillations are found for the same flow case *P95-I50* depicted
in Figure 15. The LCOs, despite having quite different amplitudes, all vibrate harmonically in the 1st edgewise mode with
≈0.67Hz. Reasons for why the FM and FSI not to agree perfectly here, could be the simple estimate of damping made by a
fit, or the assumptions done in FM simulations, for instance neglecting torsion, or basing the static deflection on the *P100-I50*
flow case. The simplified damping model fit is likely the reason for the FSI simulations at high amplitudes for *P90-I50* and





*P95-I50* not being at precisely 0W power despite being at a stable LCO level. For the *P90-I50* flow case, the conclusions are
similar, but with better agreement between intersection points and FSI results for low amplitudes.

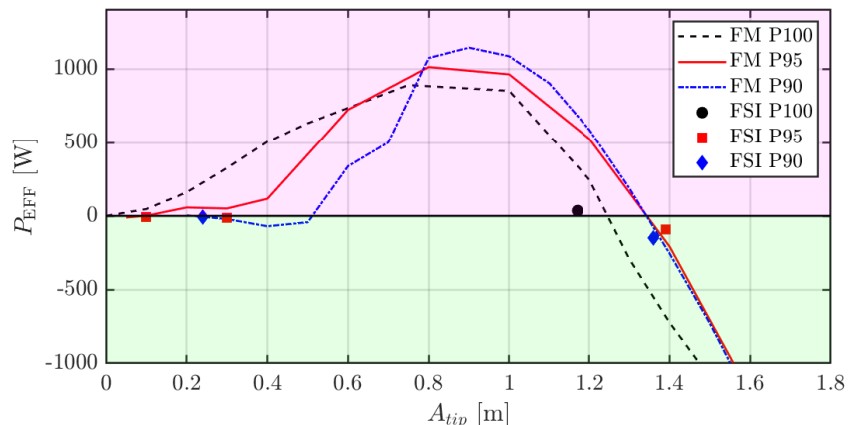

**Figure 14.** Effective power $P_{\mathrm{EFF}} = P_{\mathrm{AERO}} - P_{\mathrm{STRUC}}$ for cases P100, P95 and P90, all with inclination I50. Magenta and green areas respectively indicate unstable and stable regions. $P_{\mathrm{STRUC}}$ is here defined as eq. (4), which is why FSI results are not exactly at 0W despite being at stable LCO.

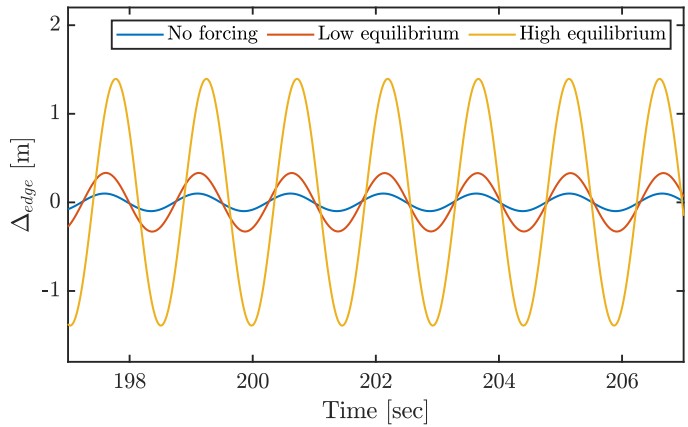

**Figure 15.** Edgewise tip displacement of flow case *P95-I50*, with three equilibrium states.

### 3.3    Time series generation of vibrations

By knowing the relation between aerodynamically injected power, structurally dissipated power and the amplitude of the mode, one can generate a time series of the development of vibration. This can be useful in terms of assessing whether a vibration





is worrisome or not, as a long development time will need long periods of steady inflow conditions, while quickly developing
vibrations can happen over a short period, making the risk here more critical.

Considering only small relative deflections in the first edgewise mode, the harmonic motion of the damped linear one degree
of freedom system in free vibration can be described as:

$$x(t) = A_0 \cdot e^{-\zeta\omega t} \cdot \sin(\tilde{\omega}t + \psi) = A(t) \cdot \sin(\tilde{\omega}t + \psi), \tag{5}$$

with the initial amplitude $A_0$, the damping ratio $\zeta$, and the phase $\psi$, which is assumed to be zero in the following. Further,
for the small damping of the first edgewise mode, the damped frequency is almost equal to the undamped natural frequency
$\omega_{1e} = \tilde{\omega} \approx \omega$. The time derivative of the amplitude can be found to be a function of the amplitude itself, the damping ratio and
the natural frequency:

$$\frac{dA}{dt} = -\zeta\omega A_0 \cdot e^{-\zeta\omega t} = -\zeta\omega A(t) \tag{6}$$

For a damped free vibration the only power dissipation is caused by the structural damping, with the power proportional to the
damping ratio $\zeta$. For the vibrations considered in this article, the aerodynamic forces are another source of power injection,
resulting in the effective power

$$P_{EFF}(A) = P_{AERO}(A) - P_{STRUC}(A) = P_{AERO}(A) - F_{STRUC} \cdot A^2, \tag{7}$$

with the assumption again that the frequency and modeshape are approximately those of the first edgewise mode. From this
assumption follows that the proportionality constant between the structural power and the structural damping ratio has the same
magnitude as the proportionality constant between the effective power and an effective damping ratio $\zeta_{eff}$:

$$\frac{\zeta_{EFF}(A)}{P_{EFF}(A)} = -\frac{\zeta_{STRUC}}{P_{STRUC}(A)} \tag{8}$$

$$\zeta_{EFF}(A) = -P_{EFF}(A) \cdot \frac{\zeta_{STRUC}}{P_{STRUC}(A)} \tag{9}$$

With this effective damping ratio, the change in tip amplitude at a specific time step can be found as

$$\frac{dA}{dt} = -\zeta_{EFF}(A(t))\omega_{1e}A(t) = P_{EFF}(A) \cdot \frac{\zeta_{STRUC}}{P_{STRUC}(A)}\omega_{1e}A(t) \tag{10}$$

With these assumptions, a simple backwards differencing scheme can be used to find the amplitude variation over time using
interpolated values of aerodynamic power found in FM simulations: The amplitude is initialized with a certain value $A_0$, the
time derivative of the amplitude is determined from Equation (10), with the effective power as shown in Figure 14, and the
amplitude in the following time step is found as $A(t + \Delta t) = A(t) + (dA/dt)\Delta t$. The amplitude time series A(t) multiplied
with $\sin(\omega_{1e}t)$ will generate a time signal of the vibration development. The fact that the aerodynamic power is obtained from
interpolation between a finite number of FM simulations, is naturally a limitation of the method, as the amplitude discretization
will determine the accuracy of the method.

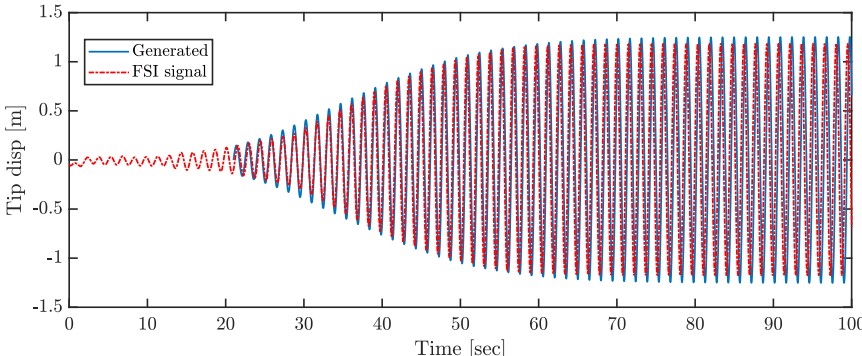

**Figure 16.** Time series of tip displacement in edgewise for FSI simulation, and generated signal based on FSI amplitude at 20 sec as initial condition. Flow case *P100-I50*

Figure 16 shows the generated signal along with the obtained FSI signal for flow case *P100-I50*. The time evolution of the vibration amplitude is in good agreement between the two signals, despite the discretization of the aerodynamic power in the given range is based on only seven FM simulations.

The two-amplitude case of *P90-I50* can also be generated, since FM simulations here also predict two LCO amplitudes, as also depicted in Fig. 17. This, by defining an initial amplitude below or above the stable region depicted in Fig. 14. Since the time generation method is based on averaged power per amplitude, from FM simulations with steady inflow, the generation will not be able to capture triggering of amplitude branches by for instance turbulence buffeting.

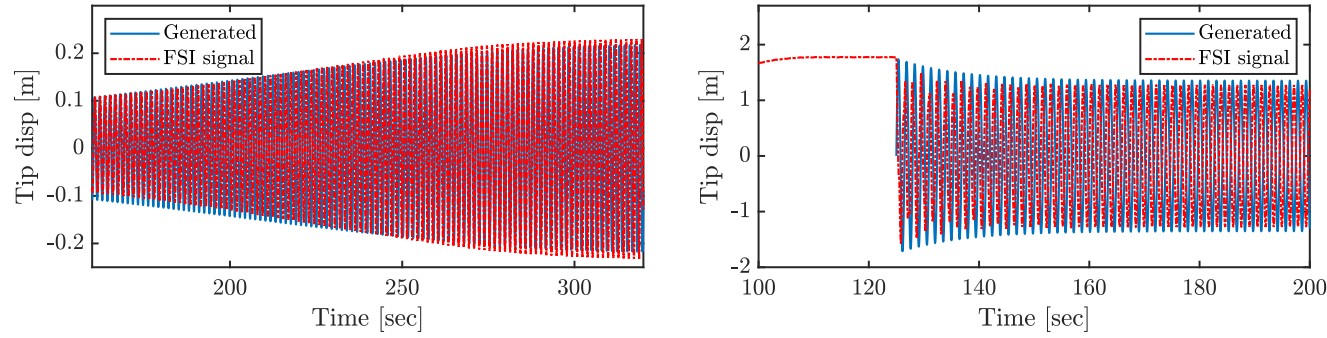

**Figure 17.** Generated time series for *P90-I50* along with corresponding FSI simulations, for low LCO amplitude (left) and high LCO amplitude (right). In FSI, the high amplitude LCO was obtained by pulling the blade to 1.75m edgewise amplitude and releasing it to vibrate in the flow.

For *P95-I50*, the FM simulations predict a slightly higher aerodynamic power than damping for the low amplitudes, meaning
that here, the generation will predict only one LCO stabilized amplitude, whereas the FSI simulation as mentioned predicted





three. This might not be a critical point, as the low amplitude LCOs are of less importance when designing the wind turbine blade.

## 4    Concluding remarks

The main objective of the current study has been to conduct and compare simulations of vortex induced vibrations (VIVs) by the
use of two methodologies both based on CFD. Simulations fully coupled to a structural solver are compared with simulations with the blade motion imposed by an analytical expression of the 1st edgewise mode instead. Fully coupled simulations are beneficial as no assumptions are made of the structural response, however limited by the fact that long time series are needed in order to reach a limit cycle oscillation (LCO). Forced motion simulations, on the other hand, have very limited initial transient as the LCO is prescribed and only the flow needs to converge to a periodic state. However, these simulations need assumptions
on the motion shape and frequency, and various amplitudes need to be simulated.

It is found for the studied cases of high inclination angles and pitch angles close to perpendicular to the chord, that a very good agreement is seen between coupled and forced motion simulations. This is found by comparing power injection along the span of the blade, which is a parameter sensitive to both loads, motion and flow shedding frequencies, making it difficult to predict.

By fitting a simple model for the blade structural damping for the 1st edgewise modeshape, it is possible not only to make good assessments of the final LCO amplitude, but also estimate the development of the motion by using data from multiple forced motion simulations with varying amplitude.

Finally, the study reveals a VIV dependency on the initial conditions of the blade motion. Cases, which at steady inflow and without initial motion do not build up VIVs, might build up large VIVs if perturbed by initial motion or unsteady loading.
Examples of such perturbations could be turbulent inflow, or actions from the turbine controller such as yawing or pitching. An example of this was demonstrated for the flow case *P95-I50*, where three different stable LCO amplitudes can be found if the initial amplitudes are varied in the FSI simulations.

## 5    Future studies

Vortex induced vibrations for wind turbines, is still an understudied field, with many possible discoveries to be made. Within
single blade VIVs, the studied parameter space can be expanded by looking at different modes, a broader range of inflow angles and velocities, inflow turbulence and much more. However, expanding the parameter space rapidly increases computational efforts needed, why the forced motion approach, examined in the current study, might become relevant in order to reduce the cost.

Looking further, investigations of full rotors and entire wind turbine setups are naturally relevant. This would allow different
modes to be excited for the same wind speed range, again expanding the parameter space. Here, the blades will interact in



terms of injection and dissipation of power, depending on their positions, and the flow might be altered by upstream blades for some positions.

## Acknowledgments

This work has been supported by the PRESTIGE project (J.no. 9090-00025B), granted by Innovation Fund Denmark. Computational resources were provided by the DTU Risø cluster Sophia (DTU Computing Center, 2021).

The authors would like to thank Pim Jacobs, Aqeel Ahmed, Bastien Duboc and the rest of the team from Siemens Gamesa Renewable Energy, for many fruitful discussions regarding the presented work in the PRESTIGE project

## Data availability statement

The geometry and the structural model of the considered wind turbine are publicly available. However, the structural and fluid solvers are licensed. The data that support the findings of this study are available upon reasonable request to the corresponding author.

## Author contributions

CG conducted FSI simulations, did analysis of FSI and FM simulations, and did the main writing of the paper. NNS conducted FM simulations and supported FSI analysis. GRP supported on FSI simulations and analysis and set up the time series generation method. SGH did initial FSI study and meshing and supported in the FSI analysis. All authors contributed in editing the paper.

## Competing interests

The authors declare that there are no competing interests.



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
