# Peer review of "Multiple limit cycle amplitudes in high fidelity predictions of standstill wind turbine blade vibrations"

_Wind Energy Science, 2022_

## Referee Comment (RC2)

[referee-annotated manuscript omitted]

---

## Author Comment (AC1)

**Revision answers - Reviewer 1**

Christian Grinderslev[1], Niels Nørmark Sørensen[1], Georg Raimund Pirrung[1], and Sergio González Horcas[1]

[1]Department of Wind Energy, Technical University of Denmark, Risø Campus, 4000, Roskilde

**Correspondence:** Christian Grinderslev (cgrinde@dtu.dk)

*First and foremost the authors would like to thank the reviewer for the good comments, questions and suggestions. They will add good value to the paper. In the following, bold text represents the original comment by the reviewer and the following italic text represents the answers from the authors.*

**The manuscript investigates if a forced motion approach is a good complementary analysis tool to fully-coupled**

5 **analyses for the study of wind turbine blade VIVs. The study compares both approaches for a 10MW wind turbine blade, with a focus on the impact of the structural damping and the initial conditions. The manuscript is well written and of high relevance to the field.**

**I have the following questions/clarifications on some of the results.**

**- Fig 6. The figure shows the power injected by the motion for different motion amplitudes. The authors point at**

10 **large power injections beyond amplitudes of 1m. However, in certain cases, large power injections are also apparent at smaller amplitudes (with power injection values larger than close to the max amplitude of 1.6m). Isn't the positive value of power more important than the amplitude value? (i.e. large P at A=0.6m could be more problematic than smaller P at A=1.6m). This could be clarified in the text.**

*The power can indeed be high for low amplitudes, but it is the opinion of the authors, that the most important aspect is the*

15 *amplitude value at the equilibrium between structural damping dissipation and aerodynamic power injection,. This will show the LCO amplitude, which the turbine blade might need to withstand for many cycles, which could lead to high fatigue damage. The moments will be highest at this point, and not so much for the lower amplitudes. The peaks of power are, however, still of interest, especially when also considering the structural damping dissipation. The difference between these two ($P_{EFF}$ in eq. (7)) will determine the vibration development speed as also described in the proposed time series generation model. This is of*

20 *interest since it is relevant to see whether the flow scenario will take little or long time to develop the VIVs, since there will be higher probability for the rapidly developing cases.*

**- Fig 9. There is an overall good agreement between FM and FSI values of Pcycle over the blade radius. Can the authors comment (also based on previous studies) on how much local positive values of Pcycle can trigger VIV by propagating throughout the whole blade? Following this, could some of the differences observed between FSI and FM**

25 **be an issue for predicting VIV occurrence based on FM simulations?**

*The presence of local positive values of power input will not be enough to introduce VIVs for this application. Since the blade vibrates based on aeroelastic modes (that in this case corresponded to the natural ones), a total positive accumulated*

*power for each of their frequencies would be needed. This can indeed seem counter-intuitive as this means that the position of the positive and negative regions spanwise have no effect, as long as the total power is positive. However, the point made by the reviewer concerning the propagation of the positive power regions is definitely a crucial aspect of the problem. While the spanwise distributions shown in Figure 9 of the manuscript do correspond to stabilized values of a constant amplitude of motion (both for FM and for FSI), the group has studied the onset of vibrations in another publication [Horcas et al 2022 https://doi.org/10.1063/5.0088036]. The results of the aforementioned work revealed that the stabilized distribution of the power could be the result of the local excitation of normal shedding, which is later on modulated by the effects of oblique shedding. As this process seems to be inherently dynamic, this observation links to the second remark made by the reviewer. Indeed, the FM simulations could be seen as quasi-steady evaluations of the exchanged power for a series of imposed amplitudes. In that way, some of the dynamic effects related to the onset of the vibrations could be overlooked. Nevertheless, the good correspondence between the generated signals that are shown in the present work (Figure 16 and Figure 17) seem to indicate that this is not the case for the targeted application. For faster developing vibrations, this match between the generated time series and FSI would likely not be as good.*

*We would like to add that in our opinion the good correspondence between FM and FSI is also related to the simplicity of the structural response shown in the FSI cases. This simplicity relies on two factors. On the one hand, the observed aeroelastic deflection shapes were found to be very similar to the purely structural mode shapes (so that the FM method can simply take the results of a modal analysis as an input). On the other hand, the blade motion was found to be dominated by a single mode. We believe that the study of co-existing modes of vibration through the FM method would complexify the application of the FM method.*

*In our findings, we have not seen cases where the FSI and FM methods disagreed significantly in the VIV occurrence, but it is worth noting that our CFD setups in both cases are more or less identical. We have, however, seen that the results obtained can be very sensitive to chosen turbulence models, numerical schemes etc. This is something we plan on publishing soon.*

**- Based on the results of this paper, it would be interesting to investigate a wider parameter space for VIV under forced motion. Is that planned as future work?**

*This is indeed planned and already being conducted in the ongoing PRESTIGE project. We plan to publish more detailed FM simulations and parametric investigations using various FM setups, looking especially at sensitivities to solver settings as well as for various flow scenarios.*

**Revision answers - Reviewer 2**

Christian Grinderslev[1], Niels Nørmark Sørensen[1], Georg Raimund Pirrung[1], and Sergio González Horcas[1]

[1]Department of Wind Energy, Technical University of Denmark, Risø Campus, 4000, Roskilde

**Correspondence:** Christian Grinderslev (cgrinde@dtu.dk)

*First and foremost the authors would like to expres our gratitude to the reviewer Vasilis A. Riziotis for the nice words and very useful comments.*

*In the following, bold text represents the original comment by the reviewer and the following italic text represents the answers from the authors.*

5 **In the paper, the authors attempt to correlate the predictions of advanced CFD, imposed motion simulations with those of fully coupled CFD aeroelastic analyses. One of the aims of this work is to allow prediction of the limit cycle amplitudes of a rotor blade in lock-in conditions, based on forced motion analyses, instead of (if possible) the computationally expensive fully coupled FSI analyses. The paper is well prepared and the work is extremely relevant for the wind energy community. I don't have much to say about the paper. I read it with great interest and I found the conclusions**

10 **important and the whole work worth publishing. My only concern is that full wind turbine aeroelastic analyses indicate that the coupling of the rotor with the rest of the turbine (tower etc.) significantly alters the coupled out-of-plane modes (in terms of frequency and damping) of the full system compared to the isolated blade modes. However, the authors acknowledge the above mentioned point in their conclusions section and they add that the only limitation for performing a full rotor analysis is computational cost which sounds reasonable. Given the above I recommend publication of the**

15 **paper with minor revision. Some corrections/comments have been added in the accompanying pdf.**

*Thank you very much for your interest and recommendation. We agree that looking at full rotor/turbine studies is of high relevance, as the modes change and a larger structural damping is present. We are at the moment looking into simulations of the full rotor in the CFD setup while have the entire wind turbine in the structural representation. This, we hope to publish in a following paper.*

20 *Thank you also very much for the specific comments in the paper. We've adjusted the text to your suggestions.*

Specific comments to paper in supplement:

Line 145:

**Why not applying a linear interpolation to the nodal values of the HAWCSTAB2 mode prediction? That would be more accurate than any global interpolation.**

25 *This is a practical matter in the CFD solver, which takes analytical motions as input. It would for sure make sense to, as you state, directly interpolate the modeshape found in HAWCSTAB2, but for practical reasons it was chosen to do an analytical fit.*

Line 196:

**Why do you think that torsion is more important than the contribution of higher frequency modes, which is omitted in the FM analysis.**

30    *This is also a good point, as we do have assumptions on the mode in FM being only the one of the first edgewise. In the presented cases, the PSDs of the structural responses from FSI show more than five orders of magnitude difference between the first edgewise to the next higher amplitude peaks, indicating that the assumption of pure first edgewise mode is good. In other cases where more natural shedding is present (e.g. for lower inclinations), this would likely be of more concern. The omission of torsion in the forced motion cases should likewise be of minor importance since the angle of attack is in a range where the*

35    *lift and drag show low dependence. This is also evident in the paper by S.G. Horcas et al, 2022, see Figures 4 and 14. For the specific cases the torsion at the tip of the blade was approximately $\pm 0.5°$ in the FSI simulations.*

[revised manuscript text omitted]